# Chronic Limb-Threatening Ischemia and the Need for Revascularization

**DOI:** 10.3390/jcm12072682

**Published:** 2023-04-04

**Authors:** Raffaella Berchiolli, Giulia Bertagna, Daniele Adami, Francesco Canovaro, Lorenzo Torri, Nicola Troisi

**Affiliations:** Vascular Surgery Unit, Department of Translational Research and New Technologies in Medicine and Surgery, University of Pisa, 56126 Pisa, Italy; raffaella.berchiolli@unipi.it (R.B.);

**Keywords:** chronic limb-threatening ischemia, peripheral arterial disease, lower-limb revascularization, peripheral bypass, endovascular treatment

## Abstract

Background: Patients presenting with critical limb-threatening ischemia (CLTI) have been increasing in number over the years. They represent a high-risk population, especially in terms of major amputation and mortality. Despite multiple guidelines concerning their management, it continues to be challenging. Decision-making between surgical and endovascular procedures should be well established, but there is still a lack of consensus concerning the best treatment strategy. The aim of this manuscript is to offer an overview of the contemporary management of CLTI patients, with a focus on the concept that evidence-based revascularization (EBR) could help surgeons to provide more appropriate treatment, avoiding improper procedures, as well as too-high-risk ones. Methods: We performed a search on MEDLINE, Embase, and Scopus from 1 January 1995 to 31 December 2022 and reviewed Global and ESVS Guidelines. A total of 150 articles were screened, but only those of high quality were considered and included in a narrative synthesis. Results: Global Vascular Guidelines have improved and standardized the way to classify and manage CLTI patients with evidence-based revascularization (EBR). Nevertheless, considering that not all patients are suitable for revascularization, a key strategy could be to stratify unfit patients by considering both clinical and non-clinical risk factors, in accordance with the concept of individual residual risk for every patient. The recent BEST-CLI trial established the superiority of autologous vein bypass graft over endovascular therapy for the revascularization of CLTI patients. However, no-option CLTI patients still represent a critical issue. Conclusions: The surgeon’s experience and skillfulness are the cornerstones of treatment and of a multidisciplinary approach. The recent BEST-CLI trial established that open surgical peripheral vascular surgery could guarantee better outcomes than the less invasive endovascular approach.

## 1. Introduction

### 1.1. Background and Definitions

Peripheral artery disease (PAD) is defined as the development of chronic arterial occlusive disease of the lower extremities due to arteriosclerosis. The most severe form of PAD is critical limb ischemia (CLI) [1]. The latter is an outdated concept that does not include the whole spectrum of signs and symptoms of patients with PAD and ischemic symptoms.

In 1982, Jamieson [2], and then other authors in the following years [3], first described CLI as ischemic rest pain with ankle pressure (AP) < 40 mmHg or tissue necrosis with AP < 60 mmHg. In their paper, the authors underlined the concept that diabetic patients should be excluded from this definition and considered a separate category to easily compare results in non-diabetic patients. The term CLI has been inadequately used for more than four decades. Furthermore, it does not include patients with different types of ischemia, leading to poor wound healing and a high risk of limb loss [4,5]. To clarify, these patients are those who may have relatively normal hemodynamic tests but still suffer from wounds as a result of diminished local perfusion (angiosomal ischemia due to the lack of adequate collateral flow, as in diabetic patients).

Over time, a lot of different classification systems concerning wound and diabetic foot ulcers (DFUs) have been proposed in order to better describe the large spectrum of signs due to this disease. Some of them are still present in our daily practice, such as the Fontaine [6] and Rutherford [7] classification systems. To overcome the extreme variability of classifications, in 2014, the Society for Vascular Surgery (SVS) created “The Lower Extremity Threatened Limb Classification System” [8]. The proposed risk stratification system is based on three main factors that influence amputation risk and clinical management: wound, ischemia, and foot infection (WIfI). Several studies have demonstrated the high clinical value of this classification; in particular, a strong correlation was demonstrated between the WIfI score, 1-year amputation-free survival, wound healing, and the need for limb revascularization [9,10,11,12]. With the advent of this classification system associated with continuous improvement in the innovations in PAD treatment, previous classifications rapidly became obsolete, and so did the previous hemodynamic cut-off of AP to define limb-threatening ischemia [8].

For this reason, in 2019, the European Society of Vascular Surgery (ESVS) released “Global Vascular Guidelines on the Management of Chronic Limb-Threatening Ischemia”, where the new term, “Chronic Limb-Threatening Ischemia” (CLTI), was described. This new definition overwhelmed all previous concepts. Indeed, it includes a wide spectrum of patients with ischemia, ranging from rest pain to extensive gangrene with increasing amputation risk. For the diagnosis of CLTI, an established PAD in association with ischemic rest pain or tissue loss is required. Pain should be present for more than two weeks and associated with at least one abnormal hemodynamic parameter, such as ankle brachial index (ABI) < 0.4, absolute AP < 50 mmHg, absolute toe pressure (TP) < 30 mmHg, transcutaneous pressure of oxygen (TcPO2) < 30 mmHg and flat or minimal pulsatile volume recording (PVR) waveforms [13]. Furthermore, to aid clinical decision-making in everyday practice, Global Guidelines propose a three-step integrated approach based on patient risk estimation, limb staging, and anatomic pattern of disease (PLAN). The first item provides the patient assessment of candidacy for limb salvage, periprocedural risk, and life expectancy. It should be performed using multiple risk stratification tools providing objective criteria. The second item is assessable using the SVS Threatened Limb Classification System (WIfI), which defines the clinical severity of ischemia. Eventually, Global Limb Anatomic Staging System (GLASS) should be used to define the overall pattern and severity of disease in the limb.

### 1.2. Epidemiology and Risk Factors for CLTI

The prevalence of PAD has been increasing in recent years, probably due to the growing prevalence of diabetes mellitus (DM) associated with the aging population. It is estimated that >200 million people have PAD worldwide, with a spectrum of symptoms ranging from none to tissue loss [14]. PAD is uncommon before the age of 50, but its rate dramatically increases with age, up to a rate of 29.4% at age > 80 years. Men have been reported to present a higher prevalence in high-income countries. In addition, PAD seems to be more prevalent among black individuals than among white individuals.

The risk factors for PAD are already well established. They can be divided into traditional and non-traditional risk factors. The first group includes older age, smoking, DM, hypertension, hypercholesterolemia, and air pollution. The link between high body mass index (BMI) and PAD is inconsistent because of controversial studies. A recent narrative review [15] explained that nutrition and diet are possible risk factors; their modification led to decreased incidence of PAD, as well as the subsequent development of major adverse cardiovascular events (MACEs) and major adverse limb events (MALEs). In addition, chronic kidney disease (CKD) is a strong risk factor for PAD and limb loss, above all in association with DM [13]. The fact that PAD is frequently undiagnosed and untreated, especially in early stages and in diabetic patients, underlines the necessity of early diagnosis and prognosis factors. Many studies in recent years have been investigating these non-traditional risk factors in order to quantify the residual risk in the PAD population. Non-traditional risk factors can be divided into clinical and non-clinical.

Among the clinical factors, sarcopenia is one of the most investigated due to its high prevalence in patients undergoing vascular surgery. It is well known that this condition is associated with adverse outcomes after vascular surgery. Several studies have suggested that low skeletal muscle (SM) areas can have an impact on PAD patient outcomes [16]. The available evidence is heterogeneous, and defining the prognostic role of sarcopenia in PAD patients is still challenging. However, lower SM area and mass are associated with higher mortality in patients suffering from PAD [17].

Another relevant risk factor to take into consideration is glycemic variability (GV). It is simply calculated with fasting plasma glucose (FPG) and HbA1c level dosage. Glycemic fluctuation and chronic hyperglycemia can trigger an inflammatory response, and GV has adverse effects on autonomic function and increases thrombogenicity, leading to the development of macrovascular disease. A cohort study [18] on 45 436 patients with prevalent type 2 diabetes investigated the relationship between GV and the occurrence of major adverse limb events (MALEs), and the impacts of GV on major adverse cardiovascular events (MACEs) in patients with diabetes. The study concluded that in patients with diabetes, higher GV led to a significantly increased risk of MALEs compared with lower GV, largely driven by the increased development of PAD and CLI. Patients with increased GV were also associated with increased risks of MACE development and death from any cause.

In the context of the lack of punctual biomarkers for assessing PAD, inflammation and remodeling in the atherosclerotic pathway assume key roles as non-clinical factors. A recent review [19] describes the possibility to build prediction models to refine PAD assessment and evaluate this multifactorial disease in detail. The circulating concentrations of some cytokines (C reactive protein (CRP) or Interleukin (IL)-6), coagulation factors (D-dimer or fibrinogen), proteases (matrix metalloproteinases (MMPs) and their inhibitors, tissue inhibitors of metalloproteinases (TIMPs)) or cardiac damage markers have been reported to be increased in PAD patients. Recently, the high-throughput sequencing of miRNAs in peripheral blood cells from patients with PAD revealed 29 differentially expressed miRNAs predicted to target protein-coding genes involved in pathologies of atherosclerotic etiology [20]. Moreover, further studies are necessary to confirm these promising results in the genomic field. In addition, Kremers et al. [21] underlined the usefulness of high-sensitivity CRP, neutrophil–lymphocyte ratio, NT-proBNP, and high-sensitivity cTnT, which seemed to be more feasible also in common laboratories, because they only involve blood samples. Combining these markers for individual risk stratification may lead to improved treatment choices and increased effectiveness of current treatment strategies.

People from low-income countries seem to present a higher prevalence of intermittent claudication (IC) and CLTI, due to the major exposure to all these risk factors.

The heterogeneity of data about the prevalence of CLTI continues to be an issue. However, CLTI includes about 10% of all PAD patients. In addition, CLTI patients are at high risk of death. Reinecke et al. [22] demonstrated a high amputation rate in patients with CLTI; in particular, patients with major tissue loss (Rutherford class 6) have a risk of limb loss up to 67.3%. Furthermore, Global Vascular Guidelines reported 4-year mortality rates of 18.9% for patients in Rutherford classes 1–3, 37.7% for patients in Rutherford class 4, 52.2% for patients in Rutherford class 5, and 63.5% for patients in Rutherford class 6.

Despite the unquestionable advances in risk factor management, best medical therapy (BMT) prescription, and modern treatments, plenty of evidence underlines the worldwide social and economic impacts of CLTI in the modern age. As already mentioned, patients with PAD may present a wide range of symptoms, ranging from claudication to extensive necrosis or gangrene. Most of them require hospitalization for surgical or endovascular interventions, while others need frequent outpatient visits to assess the stability or progression of the disease or need dressing cycles for non-healing ulcers. It has been calculated that the rate of hospitalization for PAD in 2014 in the USA was 89.5/100,000, with 137,050 (or 45%) of these having presented high-grade disease. For a mean hospital stay of 5 days, the cost was USD 15,755, resulting in an annual cost burden for the hospitalization of patients with PAD of ∼USD 6.31 billion [23]. The direct costs associated with PAD are higher than those associated with cardiovascular disease because of the polyvascular feature of the pathology and the higher number of annual cardiovascular events and hospitalization rates. In addition to direct costs, PAD may lead to large morbidity- and mortality-related productivity costs. The 2010 National Health and Wellness Surveys developed in the USA and Europe reported significant impairment in work in patients with PAD, in particular, absenteeism, presenteeism, overall work productivity loss, and activity impairment [24]. This fact underlines the need for strict risk factor control and the correct use of guideline-recommended drugs.

Despite the large improvement in risk factor control and medical treatment, the number of PAD patients who need revascularization continues to be high. The estimation of life expectancy and operative risk plays a central role in evidence-based revascularization (EBR). A lot of models have been developed over the years to stratify the risk of these patients. The existing risk models have demonstrated modest predictive abilities; indeed, patients apparently similar in overall risk, comorbidities, and clinical features have shown significant differences in terms of outcomes after revascularization. This lack of predictive ability is due to the heterogeneous nature of these models regarding predictor variables and assessed outcomes. Firstly, the inclusion of endovascular therapies is not uniform, and neither is the evaluation of the severity of foot necrosis or the adequacy of the management of medical risk factors and other risk factors. Secondly, all of the scoring systems are difficult to generalize to the entire population due to the lack of rigorous external validation. Additionally, another bias is due to indication, because many of the derivation sets only include infrainguinal bypasses or angioplasties, but not both. Eventually, with the acquisition of newer intraoperative and postoperative predictors of outcomes in CLI, these systems will become progressively complex [25].

## 2. Diagnosis

Several factors contribute to CLTI development; ischemia is not always an isolated cause of CLTI. Therefore, in patients with high suspicion of CLTI, diagnostic assessment is the first step in order to reach successful revascularization with limb salvage. 

A complete evaluation of patients with CLTI should include a physical examination, noninvasive hemodynamic tests, and imaging.

Diagnosis is critical to the management of PAD. More importantly, it appears that the patient’s dynamic assessment, functional status, and limitations in daily activities are directly associated with the outcomes. Measurements of lower-limb muscle mass, walking tests, and quality of life questionnaires are promising tools for improving decision-making and risk stratification in patients with PAD.

### 2.1. Immediate Diagnosis of PAD

Patient evaluation begins with a physical examination. The palpation of lower-limb pulses from the groin to the foot (femoral, popliteal, pedis, and posterior tibial) is useful as a bedside approach to suspected PAD. However, pulse palpation may not be effective in diagnosing and assessing the severity of PAD. 

Some undefined signs, such as cyanosis, coldness, dry skin, muscle atrophy, and dystrophic toenails are very common in PAD patients. Buerger’s test [26] is usually positive in CLTI patients; in addition, capillary refill time usually exceeds 5 s, especially in patients in the supine position or with a leg in elevation.

Particular attention should be paid to patients with CLTI and DM. They usually have sensory, motor, and autonomic neuropathy ranging from the lack of symptoms to burning pain or weakness in the feet, until the development of diabetic foot syndrome [27] in the final stages of the disease. In patients with ulcers, a probe-to-bone test is usually mandatory in order to assess its depth and detect underlying osteomyelitis.

### 2.2. Noninvasive Hemodynamic Tests

Current guidelines recommend the use of the following noninvasive hemodynamic tests in order to collect objective parameters to define the degree of ischemia and to subsequently establish the correct WIfI score:-AP;-ABI;-TP;-TcPO2;-TBI (toe–brachial index).

In addition, continuous-wave Doppler (CWD) should be cited, due to the possibility to exclude PAD (loss of triphasic pattern) with a simple handheld continuous-wave device at the bedside, which is particularly useful in diabetic patients [28,29]. Despite the noninvasiveness and availability of these tests, they should be carefully used. Especially in patients affected by DM or end-stage renal disease (ESRD), who have calcified and incompressible peripheral arteries, the measurement of AP and ABI could lead to abnormally elevated levels. This is the reason why a combination of tests is necessary for diseases including femoropopliteal and infrapopliteal arteries [30]. TP and TBI should be performed if abnormal elevated AP and ABI have been registered. Recent studies confirmed the better sensitivity of TBI (Figure 1), especially in “challenging populations”, such as those revealing high-grade calcifications [31]. Furthermore, both TP and TBI appear to be associated with cardiovascular and overall mortality, and amputation-free survival (AFS) in patients with PAD-presenting symptoms [32].

Other alternative tests, such as TcPO2, skin perfusion pressure, and plethysmography, have been used to evaluate limb perfusion, but they can be influenced by confounding factors, and they are not available in the majority of outpatient settings. Therefore, the best way to define the grade of PAD and CLTI is the combination of all these tests [33].

### 2.3. WIfI Classification System

Once complete physical and hemodynamic examinations have been performed, physicians can assess the WIfI score with the evaluation of three main factors: wound, ischemia, and foot infection. First of all, the WIfI score is useful to correlate the grade of the disease to the risk of major amputation, wound healing, and mortality. Furthermore, it is valuable to define the best treatment for patients between open and endovascular treatments [34,35]. Finally, the WIfI classification system gives physicians the possibility of restaging patients during follow-up [36].

### 2.4. Imaging

At least one vascular imaging test should be performed in all patients suffering from CLTI in order to assess the presence, extent, and severity of arterial disease; imaging should guide the decision-making about the best revascularization strategy.

Duplex ultrasound (DUS) is the first-line imaging technique in all outpatient settings thanks to its large availability and noninvasiveness. It offers the possibility to evaluate the morphological and dynamic features of the entire arterial pathway; it could be useful to plan surgical interventions [37]. A high-quality DUS test performed by well-trained operators could represent a good alternative to computed tomography angiography (CTA) in patients undergoing endovascular revascularization to minimize the use of contrast-enhanced radiological imaging [38]. Despite these advantages, a complete DUS evaluation could be a time-consuming and highly operator-dependent procedure. The assessment of the “inflow” (iliac axis) and the infrapopliteal (IP) status can sometimes be challenging (due to obesity and calcifications).

Nowadays, preoperative CTA imaging is required in almost all cases, especially when complex invasive interventions require a complete overview of the vascular bed. CTA has a sensitivity of 95% and specificity of 96% in detecting stenosis or occlusions [39]. In addition, multi-slice CTA offers the possibility of generating high-resolution images and three-dimensional reconstructions. Some authors support the concept that multi-slice CTA can only limit the use of digital subtraction angiography (DSA) in a few selected cases as a problem-solving tool when a clinical–radiological mismatch is present [40,41]. Conversely, other authors still agree on the role of DSA as the gold standard, particularly in patients with predominant infrapopliteal disease [42,43].

Other emerging diagnostic imaging techniques are magnetic resonance angiography (MRA) and CO_2_ angiography; the first one could be useful for better evaluating pedal arteries and distal runoff, whilst the second one should be considered an additional tool in patients with allergy to contrast medium or severe CKD [44,45].

## 3. Global Limb Anatomic Staging System (GLASS)

The choice of revascularization in patients suffering from CLTI had not been standardized for many years; it had been widely based on skilled surgeons’ personal opinions and preferences. All existing classifications have been based on anatomical features, such as the location and severity of arterial lesions [46]. The extension of disease correlates with the success of revascularization [47].

Global Vascular Guidelines introduced the concept of evidence-based revascularization (EBR), providing a structured plan for decision-making and aiming to adapt inter-operator differences in the best treatment to be offered.

This structured plan is founded on three dimensions:Patient risk estimation;Limb staging;Anatomic pattern of disease.

Successful revascularization, both surgical and endovascular, should guarantee the restoration of a pulsatile in-line flow from the groin to the foot through a target arterial pathway (TAP). The latter is usually selected because it is the least diseased (or the more suitable) crural artery providing runoff to the foot [13]. Furthermore, the TAP can be selected on the basis of angiosomal distribution. Recent studies showed that the direct revascularization of the tibial vessels seems to result in improved wound healing and limb salvage rates compared with indirect revascularization, with no effects on mortality or reintervention rates [48,49]. The aorto-iliac axis (AI) and common/profunda femoral arteries are considered inflow vessels. Successful and durable revascularization should correct deficits in inflow, in particular in patients who have undergone femoro-distal revascularization.

### 3.1. Patient Risk Estimation

The first step includes a complete assessment of overall patient risks in terms of candidacy for limb salvage, perioperative risk, and life expectancy. The instrument to determine the overall patient risk is the Vascular Quality Initiative (VQI) prediction model [50,51].

The objectives of revascularization in CLTI patients are well known: first, relief of pain; second, wound healing; third, preservation of limb function. Nevertheless, not all patients are eligible for revascularization, in particular those with poor functional reserves or those who are frail. Furthermore, frailty is associated with higher mortality and amputation rates; in addition, frail patients are observed to have a mortality benefit with a less invasive approach [52]. Indeed, a comprehensive approach to the treatment of CLTI patients should include a palliative limb care option [53]. Revascularization as a palliative treatment should only be considered to improve inflow for a subsequent amputation and to relieve pain [54].

### 3.2. Limb Staging

Due to the wide spectrum of CLTI clinical presentations, the assessment of the limb stage with the GLASS classification system plays a central role. Limb staging is based on the WIfI classification [8]. According to Global Vascular Guidelines, the benefits of revascularization are linked not only to the severity of ischemia but also to the WIfI stage.

In this context, the concept of the so-called “necessary revascularization” is an emerging topic; its meaning can be easily understood by looking at Global Vascular Guidelines, which suggest high benefits of revascularization in selected categories of patients [13].

The high benefit of revascularization belongs to all symptomatic patients with severe ischemia (WIfI grade 3) unless the clinical status does not make it possible (Figure 2). Furthermore, revascularization could be beneficial in case of either advanced tissue loss or infection (WIfI stage 4 limb) and moderate ischemia (WIfI ischemia grade 1 or 2). On the contrary, patients with minor tissue loss or infection and mild or moderate ischemia could be successfully treated conservatively. Revascularization should be taken into consideration in case of poor wound healing (size not reduced by ≥50%) despite targeted care after 4–6 weeks or signs of progressive clinical deterioration [13]. Recent studies demonstrated that in patients with mild-to-moderate ischemia and tissue loss, a conservative approach achieved 87% wound healing and 90% limb salvage, with limited need for deferred revascularization (14%) [55,56]. This supports the idea to offer revascularization to patients when successful revascularization can be highly expected.

### 3.3. Anatomic Pattern of Disease

The GLASS staging system focuses on infrainguinal disease, considering common/profunda femoral arteries and the iliac axis inflow vessels. To describe the status of the vascular bed, GLASS recommends a division into three segments: femoro-popliteal (FP), infrapopliteal, and inframalleolar (IM). The latter should not be considered within the primary assessment of the limb stage. For a correct definition of the anatomic disease stage, GLASS combines each grade of the FP and IP segments, obtaining a three-stage classification that directly correlates with limb-based patency (LBP).

## 4. Evidence-Based Revascularization

For decades, open surgical revascularization has been the gold standard of treatment in CLTI patients [57,58]. The advent of new technologies and knowledge in the endovascular field has led to a steady increase in successful procedure rates using this approach, with good short-term outcomes and less resource use [59]. Global Vascular Guidelines report their recommendations mainly based on the Bypass Versus Angioplasty for Severe Ischemia of the Leg (BASIL) trial, which has represented, for many years, the only randomized control trial comparing endovascular and open surgical revascularization in patients with CLTI. The BASIL trial concluded that in patients presenting with severe limb ischemia due to infrainguinal disease, the bypass-surgery-first strategy and the balloon-angioplasty-first strategy were associated with similar results in terms of amputation-free survival at two years. Conversely, in the long-term follow-up period, after two years, open surgical bypass seemed to be associated with a significantly reduced risk of death, amputation, or both. Furthermore, patients who had been assigned to receive surgery first rather than angioplasty first seemed to be more likely to remain alive in the future [60]. However, the BASIL trial is considered not applicable to the real world due to the development of a wide spectrum of newer devices and techniques. Furthermore, the current state of evidence in CLTI remains unclear because of the lack of data about endovascular approaches to distal and complex disease patterns.

It is well known that patients suffering from CLTI have a multilevel disease, usually involving both inflow and outflow vessels. Deficit in inflow is defined in Global Vascular Guidelines. The decision of performing staged or multilevel revascularization has to be customized to the individual patient after a complete assessment of the average operative risk, the severity of ischemia, and anatomical features. However, the treatment of choice for inflow disease located on the femoral bifurcation, as well as common femoral artery disease, still remains endarterectomy, which has shown to have low perioperative complications and long-term durability [61,62]. Concerning the treatment of the AI axis, nowadays, the endovascular approach with bare metal or covered stents guarantees successful outcomes, overcoming the high surgical risk of open surgery [63,64]. The latter is almost exclusively dedicated to extensive occlusions and after the failure of endovascular procedures in average-risk patients.

Regarding the outflow status, Global Vascular Guidelines provide a table for the preferred infrainguinal revascularization procedure (open or endovascular) based on the WIfI limb stage and on GLASS stages in patients with the great saphenous vein (GSV) available as a conduit. The techniques are complementary, with a prevalence of the endovascular approach for lower anatomic complexity. Instead, open surgery should be indicated in the presence of mild-to-severe WIfI limb stage (3 or 4) and high anatomical complexity. In intermediate ranges, there is a lack of consensus about the best treatment, but a recent study demonstrated the superiority of open bypass over endovascular therapy in terms of relief from pain, wound healing, MALEs, and death [65]. In patients with lower-grade ischemia, revascularization (both open and endovascular) is not recommended [13].

All average-risk patients with CLTI that are candidates for surgical revascularization should undergo a complete DUS scan of ipsilateral GSV to assess its availability and quality. It is well known that GSV less than 3 mm in diameter could cause loss of patency and reduced AFS during the follow-up period. Despite this, many authors agree on the use of GSV between 2 and 3 mm in diameter rather than the use of other conduits [66] (Figure 3).

Obviously, there are other factors that could affect the primary patency of open bypasses, such as distal anastomosis performed on a tibial or foot vessel and the use of a suprafascial tributary collateral as a graft [67]. In cases where GSV is not available, it is reasonable to use other autologous conduits, such as small saphenous veins and brachial veins. There are some studies comparing the use of brachial or spliced vein bypass conduits with tibial angioplasty alone or single-segment GSV (S-SGSV) bypass, concluding that when GSV is not available, spliced arm vein grafts could provide durable lower-extremity revascularization with favorable patency and limb preservation rates [68,69]. Nevertheless, these types of autologous grafts have a higher risk of failure; therefore, closer surveillance is mandatory in order to guarantee primary assisted patency [70].

However, in high-risk patients with severe CLTI, plain balloon angioplasty (PBA) still remains a valid option in patients with suitable IP anatomy (Figure 4), although bypass surgery seems to give better outcomes [71]. In recent years, a lot of technical advances in endovascular interventions have been made, such as low-profile catheters and sheaths, more navigable wires, and retrograde accesses, potentially used in complex lesions located distally on the foot. Furthermore, to achieve in-line flow to the foot, the “plantar loop technique” is nowadays well-described and widely applied [72].

Despite these numerous advances in revascularization techniques and anesthetic drugs, surgical or endovascular revascularization may not be feasible in some patients, even if technically possible, because of significant comorbidities and reduced life expectancy. Patients with PAD are at increased risk of MACEs and MALEs, especially in the perioperative period. Furthermore, these events are not limited to patients with the most severe manifestations of PAD but are also seen in patients with less severe ones [73]. Therefore, the goal of EBR is also identifying those patients where revascularization would only provide increased risks of MACEs and MALEs rather than real benefits. The therapeutic choice should avoid unnecessary revascularization and should be based on the patient as a whole rather than only aiming at limb salvage. In this scenario, focusing on improved risk-modifying therapy might be the key to the future management of PAD.

### 4.1. Multidisciplinary Team Management

A single specialist does not possess all the necessary skills to manage complex patients with multiple comorbidities. For this reason, it is useful to create a team of specialists with the required skills. An intermediate model of a multidisciplinary team dedicated to the management of CLTI patients should include at least a vascular surgeon, an endocrinologist, an interventionalist, an orthopedic surgeon, a podiatric surgeon, a diabetic and wound nurse, a physical therapist, a diabetes educator and a nutritionist [13]. Therefore, the multidisciplinary treatment of CLTI is often defined as the combination of bypass surgery, endovascular treatment, wound healing, and rehabilitation therapy that is performed for the purpose of saving limbs and lives. These intensive and complementary therapies performed by various specialists are essential for successfully treating these patients. They can be evaluated from multiple perspectives, and the best treatment could be selected after a collective decision and opinion sharing. The impact of multidisciplinary teams has been well established; they can improve processes, time to intervention, and outcomes. Indeed, Mii et al. demonstrated that aggressive wound care performed by a multidisciplinary dedicated team shortened the time to wound healing and increased the rate of wound healing within 1 year [74]. Similarly, Zayed et al. showed that a multidisciplinary approach improved the limb salvage rate in high-risk CLTI patients [75].

In summary, the best way to treat CLTI patients, especially those with multiple comorbidities, is by offering an integrated multidisciplinary approach able to manage all the aspects of the disease.

### 4.2. BEST CLI Trial

As already mentioned, the choice between open surgery and endovascular therapy as the initial treatment varies a lot among surgeons and mostly depends on the patient’s surgical risk, arterial disease pattern, and the availability of an adequate autologous vein conduit, as well as the patient’s preference and surgeon’s preference and skills. The Best Endovascular versus Best Surgical Therapy in Patients with CLTI (BEST-CLI) trial was developed to determine if endovascular revascularization was superior to the surgical one in patients with CLTI and judged suitable for both procedures [76].

BEST-CLI was an international, randomized, prospective, multicenter, open-label, superiority trial. It included two parallel studies based on the preoperative assessment of the availability of an autologous vein conduit for bypass. Cohort 1 included patients with available GSV, and cohort 2, patients who needed an alternative bypass conduit.

Enrollment began in August 2014 and continued until October 2019, for a total of 1830 patients with CTLI. The primary outcome measure was a combination of major adverse limb events, defined as amputation above the ankle, a major reintervention, or death from any cause.

A total of 1434 patients with available GSV were enrolled in the cohort, with 718 receiving surgical bypass and 716 receiving endovascular therapy. Major adverse limb events or death from any cause occurred in 42.6% of the surgical group and in 57.4% of the endovascular one. In addition, major reintervention occurred in 9.2% of patients belonging to the surgical group and in 23.7% of those belonging to the endovascular group. Furthermore, above-ankle amputations occurred in 10.4% of the surgical group and in 14.9% of the endovascular one. The incidence rates of death from any cause and preoperative death were similar among the two groups. Indeed, patients in the surgical group seemed to present lower incidence rates of new or recurrent CLTI events than those in the endovascular group.

A total of 396 patients without adequate GSV were enrolled in cohort 2, with 197 receiving surgical treatment and 199 being subjected to the endovascular approach. The mean follow-up was 1.6 years. In addition, in this cohort, the baseline patient characteristics were well-balanced between the two groups. In the surgical group, 48 bypasses were performed with alternative autogenous veins, and 119 bypasses, with a prosthetic conduit.

The primary outcomes of MALEs or death from any cause occurred in 42.8% of patients belonging to the surgical group and in 47.7% of patients belonging to the endovascular one. The surgical group showed better results in terms of time until major reintervention, while no differences in time until above-ankle amputation or death from any cause between the two groups were recorded [76]. Furthermore, there were no differences between surgical and endovascular groups in the incidence of new or recurrent CLTI events.

Regarding MALEs, they occurred in 3.3% of patients from the date of randomization to 30 days after the procedure and in 31.3% of patients by the end of the trial.

The overall results of this trial suggest that preoperative planning in patients with CLTI should include patient risk assessment as well as the availability of GSV. In patients with good quality GSV, vein bypass was a superior initial strategy. Conversely, patients without adequate vein conduits benefitted from an endovascular approach. However, in the latter patients, the outcomes associated with primary endovascular intervention were not significantly different from those with initial open bypass.

### 4.3. No-Option CLTI Patients

Although the proper treatment in patients with CLTI is revascularization, unfortunately, there is a non-negligible part of patients unsuitable for revascularization for anatomical or physiological reasons. This subgroup is defined as “no-option CLTI” [77], which is a relatively new concept. It is based on the assumption of potential, successful revascularization in CLTI patients without a suitable TAP and no visible arterial circulation in the foot (desert foot). This type of arterial disease with occluded plantar arch, foot, and tibial arteries is more common in diabetic and ESRD patients, and in the past, it was the main criterion to perform a major amputation. Nowadays, the development of new technologies and devices, especially in the endovascular field, has contributed to relaunching the topic of no-option CLTI as a new field to explore before thinking about major amputations.

The real incidence and prevalence of no-option CLTI patients are not available because of the lack of epidemiological studies. However, it is well known that no-option CLTI is associated with ischemic heart disease, ESRD, and heart failure.

Therefore, the primary goals of these patients are relieving ischemic pain, healing ulcers, avoiding limb loss, improving the quality of life, and prolonging survival.

The concept of autologous cell therapy to treat no-option CLTI comes from the tumoral field [78]. In recent years, a lot of studies have been developed to evaluate the efficacy of bone marrow mononuclear cells (BM-MNCs) and peripheral marrow mononuclear cells (PM-MNCs). Their use seems to reduce the rate of major amputations and promote wound healing. In particular, the use of PM-MNCs, which consist of a heterogeneous population of lymphocytes and monocytes, seems to be the most effective autologous cell therapy due to their easy collection and efficacy in diabetic patients.

## 5. Conclusions

CLTI is not only one of the most widespread diseases and among the most challenging ones for vascular surgeons but also one of the highest impact burdens on the economic system. Global Vascular Guidelines have improved the way to manage CLTI patients with evidence-based revascularization (EBR). Revascularization itself represents non-negligible risks of MACEs and MALEs, which is the reason why accurate stratification considering all the individual risk factors, both clinical and non-clinical, and the new idea of “residual risk” for the patient is fundamental in the management of this pathology. In fact, what is detectable from the current literature is that patients apparently similar in overall risk, comorbidities, and clinical characteristics have significant differences in outcomes after the revascularization procedure. The aim of EBR is to only perform procedures where the risk–benefit ratio is well balanced, so as to avoid overtreatment and unnecessary revascularization. In the most challenging patients, the vascular team-led multidisciplinary evaluation of cases is strongly recommended together with the expertise of the surgeon. Regarding patients at average risk, the BEST-CLI trial established that open surgical peripheral vascular surgery, which guarantees better outcomes than the less invasive endovascular approach, is still the best option.

## Figures and Tables

**Figure 1 jcm-12-02682-f001:**
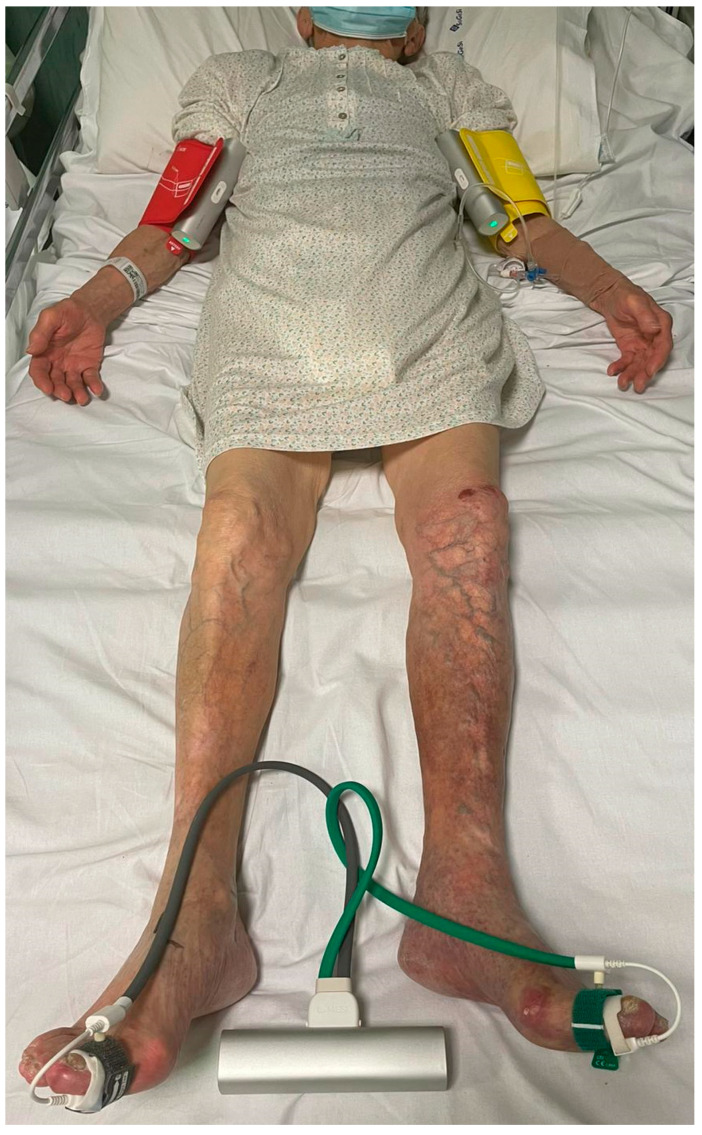
Method used to calculate TBI in patients with CLTI.

**Figure 2 jcm-12-02682-f002:**
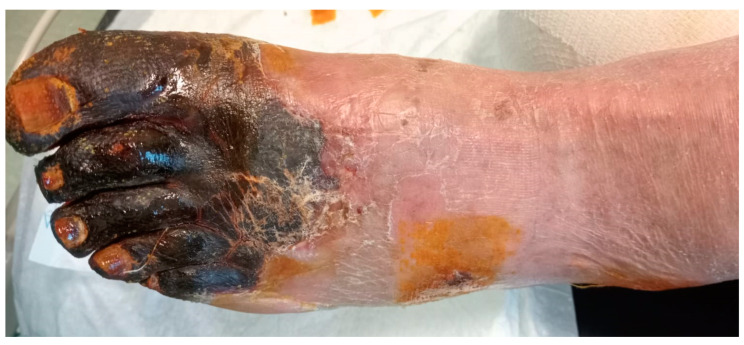
Extensive gangrene of the forefoot in patient with severe ischemia (WIfI grade 3).

**Figure 3 jcm-12-02682-f003:**
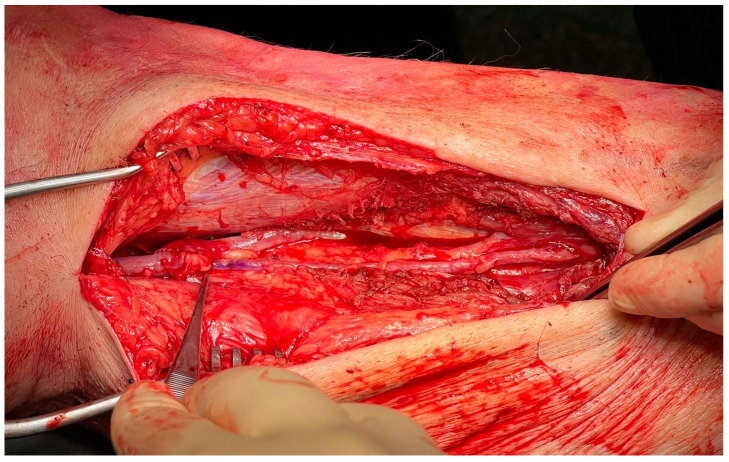
Infragenicular vein bypass in CLTI patient.

**Figure 4 jcm-12-02682-f004:**
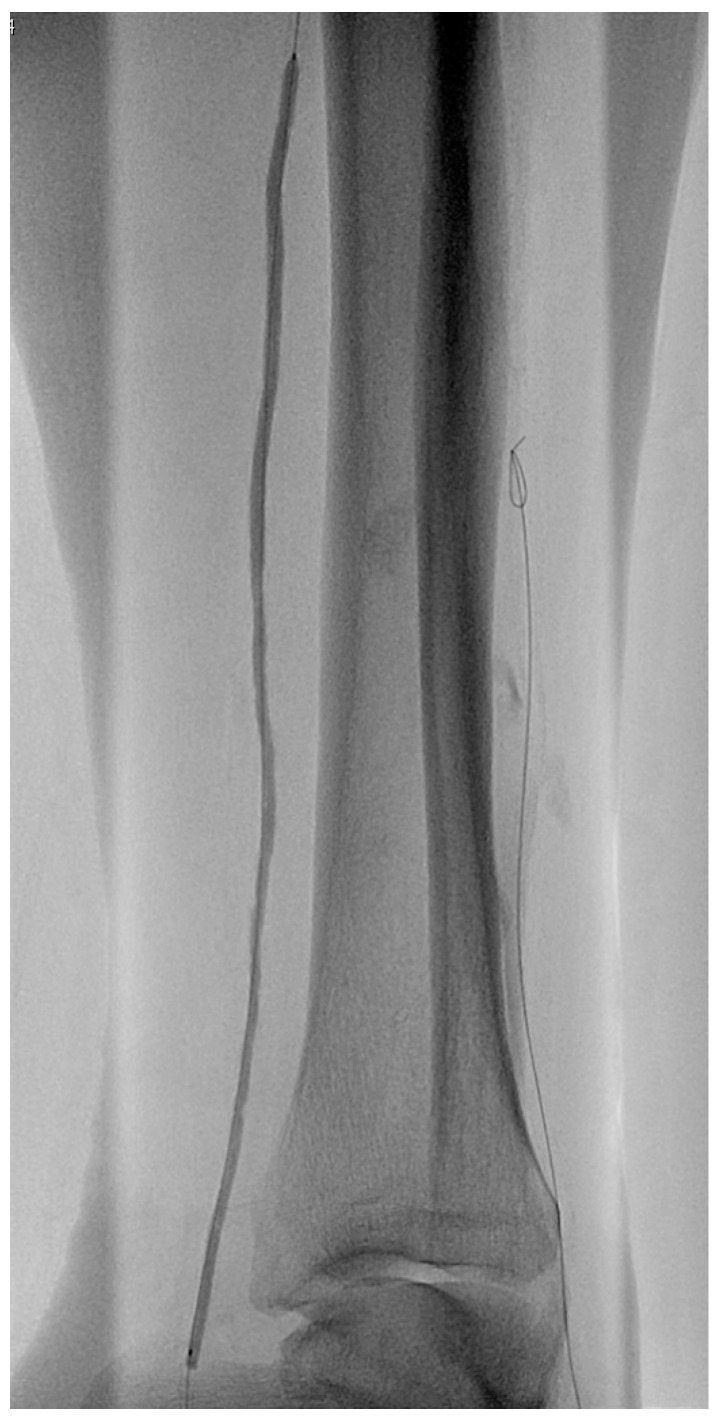
Angioplasty of posterior tibial artery after pedal plantar loop technique.

## Data Availability

Not applicable.

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
