# Peer review of "Chronic Limb-Threatening Ischemia and the Need for Revascularization"

_jcm, 2023, doi:10.3390/jcm12072682_

Round 1
Reviewer 1 Report
The authors gave a clear and informative overview of several aspects of CLTI, including recently published tools/systems such as GLASS, PLAN and TAP. Also, the landmark trials (BASIL and recently published BEST CLI) are mentioned to show the evidence in treatment of CLTI. The authors have succeeded to write it catchy/to the point.
Page 1, line 31-32
In which way is the term “CLI” inadequately used (for more than four decades)?
Page 2/3, line 94-95
The absence or presence of palpable pulsations of the ankle/foot arteries are inaccurate to rule in or rule out PAD. Nevertheless, it can give you a suspicion of the presence of PAD. Maybe you could nuance line 94-95 in your manuscript, in particular “immediate diagnosis of PAD”.
Page 3, line 105-128
You gave a clear overview of the noninvasive hemodynamic tests. It is important to note that all bedside tests have pros and cons; there is no test that has it all (ability to rule in and rule out PAD). In addition, I miss the continuous wave Doppler (CWD) (or ankle arterial waveform), as described in:
- Normahani et al 2021, A systematic review and meta-analysis of the diagnostic accuracy of point-of-care tests used to establish the presence of peripheral arterial disease in people with diabetes
- Brouwers et al 2022, Reliability of bedside tests for diagnosing peripheral arterial disease in patients prone to medial arterial calcification: A systematic review
Page 7, line 249
The BASIL trial has reference 48 (in the reference list). In the text ref 49 is mentioned regarding the BASIL trial.
Page 7, line 243-249
In the manuscript it is mentioned that the BASIL trial shows similar results in terms of amputation free survival between bypass and endovascular therapy. The BASIL trial shows indeed no significant difference regarding amputation free survival between surgery and endovascular therapy at 2 years. However, in survivors after 2 years, bypass surgery was associated with improved survival and amputation free survival. Maybe you could mention this nuance in your manuscript?
Author Response
POINT BY POINT RESPONSE TO REVIEWERS jcm-2254783
Dear Editors and Reviewers,
we would like to thank you for your time and effort in making our study better with all your valuable comments. We have done our best to incorporate all your suggestions. All the changes are highlighted in the manuscript and the responses are as stated below:
REVIEWER 2
The authors gave a clear and informative overview of several aspects of CLTI, including recently published tools/systems such as GLASS, PLAN and TAP. Also, the landmark trials (BASIL and recently published BEST CLI) are mentioned to show the evidence in treatment of CLTI. The authors have succeeded to write it catchy/to the point.
Page 1, line 31-32
In which way is the term “CLI” inadequately used (for more than four decades)?
Response: thank you for your comment
The inadequate use of the term “critical limb ischemia” is due to the fact that it doesn’t include the whole spectrum of patients evaluated and treated for threatening-limb ischemia in the modern age. This spectrum comprehends patients with different degree of ischemia which can lead to a delay in wounds healing or increase the risk of major amputation.
Page 2/3, line 94-95
The absence or presence of palpable pulsations of the ankle/foot arteries are inaccurate to rule in or rule out PAD. Nevertheless, it can give you a suspicion of the presence of PAD. Maybe you could nuance line 94-95 in your manuscript, in particular “immediate diagnosis of PAD”.
Response: thank you for your comment
Page 2/3, line 94-95- change “physical examination” to “immediate diagnosis of PAD”
Page 3, line 105-128
You gave a clear overview of the noninvasive hemodynamic tests. It is important to note that all bedside tests have pros and cons; there is no test that has it all (ability to rule in and rule out PAD). In addition, I miss the continuous wave Doppler (CWD) (or ankle arterial waveform), as described in:
- Normahani et al 2021, A systematic review and meta-analysis of the diagnostic accuracy of point-of-care tests used to establish the presence of peripheral arterial disease in people with diabetes
- Brouwers et al 2022, Reliability of bedside tests for diagnosing peripheral arterial disease in patients prone to medial arterial calcification: A systematic review
Response: thank you for your comment
Page 3, line 118-119: we modified these lines as follows, adding these two references:
“In addition, the continuous wave Doppler (CWD) should be cited, due to the possibility to exclude PAD (loose of triphasic pattern) with a simple handheld continuous wave device at the bedside, particularly useful in diabetic patients [28,29]. „
Page 7, line 249
The BASIL trial has reference 48 (in the reference list). In the text ref 49 is mentioned regarding the BASIL trial.
Response: thank you for your comment
We modified reference “48” concerning BASIL trial to “[60]”, after addition of other references suggested. We verified the correspondence between text and reference list.
- Adam DJ, Beard JD, Cleveland T, Bell J, Bradbury AW, Forbes JF, Fowkes FG, Gillepsie I, Ruckley CV, Raab G, Storkey H; BASIL trial participants. Bypass versus angioplasty in severe ischaemia of the leg (BASIL): multicentre, randomised controlled trial. Lancet 2005, 366, 1925-1934..
Page 7, line 243-249
In the manuscript it is mentioned that the BASIL trial shows similar results in terms of amputation free survival between bypass and endovascular therapy. The BASIL trial shows indeed no significant difference regarding amputation free survival between surgery and endovascular therapy at 2 years. However, in survivors after 2 years, bypass surgery was associated with improved survival and amputation free survival. Maybe you could mention this nuance in your manuscript?
Response: thank you for your comment
Page 7, line 243-249: we clarified this paragraph as follows:
“… amputation-free survival at two years. Conversely, in the long term follow-up period, after two years, open surgical bypass seemed to be associated with a significantly reduced risk of death, amputation or both. Furthermore, patients who had been assigned to receive surgery first rather than angioplasty first seemed to be more likely to remain alive in the future [60].”
Reviewer 2 Report
Major Comments
Title and content:
- The content of this review is not clearly stated in the title. The description of current knowledge could be enriched by the original contribution of the authors. Is there any gap in the literature presented that could be a source of new considerations and potentially provide new insights for the study?Please, revise the paper.
Abstract:
- The abstract lacks structure and overall logical flow, and the purpose, although vaguely described at the beginning, is not clear enough. We suggest divide the abstract into a 4-point system including background, methods/structure of the review, results (with key messages), and conclusion. Please stress the "innovative content" this document is supposed to provide.Introduction:
- The introduction section is a description of already established knowledge that may risk to appear “sterile”. We would appreciate an original contribution from the authors that can further enrich the current knowledge on CTLI management.Chapter 1, Introduction, sub-chapter 1.1 Background and Definition:
- As this is only an introduction, briefly describe/mention the SVS PLAN, a recommended approach to evaluate patients with CTLI. This brief introduction would also include the WIfI classification. - Line 32+33: the sentence is not clear An example could support the message you wanted to transmit. - Line 46+47: citation missing. This is a meaningful statement that should be supported by a bibliographic reference.Chapter 1, Introduction, sub-chapter 1.2 Epidemiology and risk factors for CLTI
- Risk factors of PAD and CTLI can be divided into two groups: traditional and non-traditional risk factors. Also, non-traditional risk factors can be divided into clinical (such as sarcopenia, glycemic variability…) and non-clinical (such as inflammatory cytokines, apolipoproteins, adipokines, miRNAs, transcription factors, serum biomarkers…) risk factors. There is a growing body of evidence on this fundamental topic that represents a key strategy to accurately stratify individual patient risk and understand the mechanism sustaining the “Residual risk” in the PAD population. Please, expand further these topics. - Describe the economic burden of PAD and PAD with CTLI. - Patients apparently similar in overall risk, comorbidities, and clinical characteristics have significant differences in outcomes after the revascularization procedure. Briefly discuss the current limitations affecting reliable population risk stratification. - Page 2, line 69: correct the citation number (14). That paper (narrative review and not a metanalysis) is an example of original contribution (impact of dietary risk factors, eating behaviours and nutritional habits on PAD and outcomes of PAD).Please correct any other mistakes in the bibliography and reference association. - Page 2, line 80-82: the latest CTLI guidance is from 2019. Did you refer to a specific topic covered in the 2015 guidelines? Otherwise correct.
Chapter 2, Diagnosis:
- Page 2, sub-chapter 2.1. Physical examination, line 94-95: pulse palpation may be useful as bed-side approach to suspect PAD. Furthermore, pulse palpation may not be effective in diagnosing and assessing the severity of PAD. - Page 4, sub-chapter 2.3. WIfI classification system, line 135-139: the comparison with TNM is not really useful for the purpose of the review and the reference added at the end of the phrase doesn't mention any of the similarities reported. - Diagnosis is critical to the management of PAD. More importantly, it appears to be the patients' dynamic assessment, functional status, and limitations to daily activities that are directly associated with outcomes. Measurements of lower limb muscle mass, walking tests, and quality of life questionnaires are promising tools for improving decision-making and risk stratification of patients with PAD.
Chapter 3, The Global Limb Anatomic Staging System (GLASS):
- sub-chapter 3.2: Limb staging, page 6, line 213-214: It could be useful to adapt the graphic referred to in the paper and add it so the reader can have a quick overview - line 185: what do you mean by saying “It is usually the most diseased crural artery providing runoff to the foot”? Please, clarify.Chapter 4, Evidence-Based Revascularization
- Line 263: common femoral artery disease is another indication for surgery as first recommended treatment. - The revascularization procedure itself is an important risk factor for MACE and MALE in PAD population. Therefore, evidence-based revascularization also means identifying all those cases where revascularization would only provide an increased risk of mortality and morbidity rather than a benefit. In fact, “unnecessary revascularization” and the support of risk-modifying therapy appear to be the real future for effective PAD management. Please discuss. - Vascular Team and multidisciplinary evaluation of cases is recommended for the most challenging patients. Please discuss the importance of shared decision making and the effects of multidisciplinary management of PAD patients.Conclusion:
- The conclusion may appear vague as it does not remark the importance of “Evidence-Based Revascularization” and “Necessary Revascularization”. Rewrite the conclusion including the revisions suggested in this peer review.Minor Comments
- Overall grammar (especially use of prepositions) should be revised - There is excessive use of passive tenses which overcomplicates the reading - Page 6, line 217: the wording is not clear - Page 10, line 349-350: missing reference - Page 10, line 364: Reference number 64 is not listed in the “References” section - Page 11, line 421: reference number 13 is overcited throughout the paperAuthor Response
POINT BY POINT RESPONSE TO REVIEWERS jcm-2254783
Dear Editors and Reviewers,
we would like to thank you for your time and effort in making our study better with all your valuable comments. We have done our best to incorporate all your suggestions. All the changes are highlighted in the manuscript and the responses are as stated below:
REVIEWER 1
Major Comments
Title and content:
- The content of this review is not clearly stated in the title. The description of current knowledge could be enriched by the original contribution of the authors. Is there any gap in the literature presented that could be a source of new considerations and potentially provide new insights for the study?
Please, revise the paper.
Abstract:
- The abstract lacks structure and overall logical flow, and the purpose, although vaguely described at the beginning, is not clear enough. We suggest divide the abstract into a 4-point system including background, methods/structure of the review, results (with key messages), and conclusion. Please stress the "innovative content" this document is supposed to provide.
Response: thank you for your comment.
We modified the structure of the abstract as follows:
“Background
Patients presenting with critical limb-threatening ischemia (CLTI) have been increasing in number during years. They represent a high risk population, especially in terms of major amputation and mortality. Despite multiple Guidelines concerning their management, it continues to be challenging. Decision-making between surgical and endovascular procedures should be well-established, but there is still a lack of consensus concerning the best treatment strategy. The aim of this manuscript is to offer an overview of contemporary management of CLTI patients, with a highlight on the concept that evidence-based revascularization (EBR) could help surgeons to be more appropriate in treatment, avoiding improper procedures, as well as too high risk ones.
Methods
We performed a systematic search on MEDLINE, Embase and Scopus from 1st January 1995 to 31st December 2022 and reviewed GLOBAL and ESVS Guidelines. A total of 150 articles were screened but only those with a high quality were considered and included in a narrative synthesis.
Results
Global Vascular Guidelines have improved and standardized the way to classify and manage CLTI patients through an evidence based revascularization (EBR). Nevertheless, considering that not all patients are suitable for revascularization, a key strategy could be to stratify unfit patients considering both clinical and non-clinical risk factors, so to keep up with the concept of individual residual risk of every patient. Surely, the recent BEST-CLI trial established the superiority of autologous vein bypass graft over the endovascular therapy for the revascularization of CLTI patients. No-option CLTI patients still represent a critical issue.
Conclusions
Surgeon's experience and skillness are the cornerstones of treatment, as well as a multidisciplinary approach. The recent BEST-CLI trial established that open surgical peripheral vascular surgery could guarantee better outcomes rather than the less invasive endovascular approach.”
and stressed the concept about the innovative content of our paper as follows:
“The aim of this manuscript is to offer an overview of contemporary management of CLTI patients, with a highlight on the concept that evidence-based revascularization (EBR) could help surgeons to be more appropriate in treatment, avoiding improper procedures, as well as too high risk ones.”
Introduction:
- The introduction section is a description of already established knowledge that may risk to appear “sterile”. We would appreciate an original contribution from the authors that can further enrich the current knowledge on CTLI management.
Response: We received an invitation to write a review article in a Special Issue "Clinical Management of Limb Ischemia" with a pre-established title „Chronic limb-threatening ischemia: the necessary revascularization“. For this reason, we cannot write an original contribution.
Chapter 1, Introduction, sub-chapter 1.1 Background and Definition:
- As this is only an introduction, briefly describe/mention the SVS PLAN, a recommended approach to evaluate patients with CTLI. This brief introduction would also include the WIfI classification.
Response: thank you for your comment
We briefly described the PLAN as suggested as follows:
“Furthermore, to aid the clinical decision-making of everyday practice, GLOBAL Guidelines propose a three steps integrated approach based on: Patients risk estimation, Limb staging and ANathomic pattern of disease (PLAN). The first item provides the patient’s assessment for candidacy of limb salvage, periprocedural risk and life expectancy. It should be performed through multiple risk stratification tools providing objective criteria. The second item is assessable using the SVS Threatened Limb Classification System (WIfI), which defines the clinical severity of ischemia. Eventually, the Global Limb Anatomic Staging System (GLASS) should be used to define the overall pattern and severity of disease in the limb.“
- Line 32+33: the sentence is not clear An example could support the message you wanted to transmit.
Response: thank you for your comment
We explained this concept as follows:
“To clarify, these patients are those who may have relatively normal hemodynamic tests, but nevertheless suffer from wounds as a result of diminished local perfusion (angiosomal ischemia due to a lack of adequate collateral flow as in diabetic patients). „
- Line 46+47: citation missing. This is a meaningful statement that should be supported by a bibliographic reference.
Response: thank you for your comment
We supported this statement by adding reference “[8]”.
- Mills JL Sr, Conte MS, Armstrong DG, Pomposelli FB, Schanzer A, Sidawy AN, Andros G; Society for Vascular Surgery Lower Extremity Guidelines Committee. The Society for Vascular Surgery Lower Extremity Threatened Limb Classification System: risk stratification based on wound, ischemia, and foot infection (WIfI). J Vasc Surg 2014, 59, 220-234.
Chapter 1, Introduction, sub-chapter 1.2 Epidemiology and risk factors for CLTI
- Risk factors of PAD and CTLI can be divided into two groups: traditional and non-traditional risk factors. Also, non-traditional risk factors can be divided into clinical (such as sarcopenia, glycemic variability…) and non-clinical (such as inflammatory cytokines, apolipoproteins, adipokines, miRNAs, transcription factors, serum biomarkers…) risk factors. There is a growing body of evidence on this fundamental topic that represents a key strategy to accurately stratify individual patient risk and understand the mechanism sustaining the “Residual risk” in the PAD population. Please, expand further these topics.
Response: thank you for your comment
We expanded the topics as follows:
„Risk factors for PAD are already well-established. They can be divided into traditional and non-traditional risk factors. The first group includes older age, smoking, DM, hypertension, hypercholesterolemia and air pollution. The link between high body mass index (BMI) and PAD is inconsistent, because of controversial studies.“
“… above all in association with DM [13]. Considering that PAD is frequently undiagnosed and untreated especially in early stages and also in diabetic patients, underlines the necessity of early diagnosis and prognosis factors. Many studies during the last years have been investigating these nontraditional risk factors in order to quantify the residual risk in PAD population. Nontraditional risk factors can be divided as well into clinical and non-clinical.
Among clinical factors sarcopenia is one of the most investigated, due to its high prevalence in patients undergoing vascular surgery. It is well known that this condition is associated with adverse outcomes after vascular surgery. Several studies have suggested that low skeletal muscles (SM) area can have an impact on PAD patients ‘outcomes [16]. Evidence available is heterogeneous and defining the prognostic role of sarcopenia in PAD patients is still challenging. However, lower SM area and mass are associated with higher mortality in patients suffering from PAD [17].
Another relevant risk factor to take into consideration is glycemic variability (GV). It is simply calculated through fasting plasma glucose (FPG) and HbA1c levels dosage. Glycemic fluctuation and chronic hyperglycemia can trigger the inflammatory response and GV has adverse effects on autonomic function and increases the thrombogenicity, leading to the development of macrovascular disease. A cohort study [18] on 45 436 patients with prevalent type 2 diabetes investigated the relationship between GV and the occurrence of major adverse limb events (MALEs) and the impacts of GV on major adverse cardio-vascular events (MACEs) in patients with diabetes. The study concluded that in patients with diabetes, higher GV led to significantly increased risks of MALEs compared with lower GV, driven largely by the increased development of PAD and CLI. Patients with increased GV were also associated with increased risks of MACEs development and death for any causes.
In a context of lack of punctual biomarkers for assessing PAD, inflammation and remodeling in the atherosclerotic pathway assume a key role as non-clinical factors. A recent review [19] particularly describes the possibility to build prediction models to refine PAD assessment and evaluate this multifactorial disease. The circulating concentrations of some cytokines (C reactive protein (CRP) or Interleukin (IL)-6), coagulation factors (D-dimer or fibrinogen), proteases (Matrix metalloproteinases (MMPs) and their inhibitors, tissue inhibitors of metalloproteinases (TIMPs)) or cardiac damage markers have been reported to be increased in PAD patients. Recently, high-throughput sequencing of miRNAs in peripheral blood cells of patients with PAD revealed 29 differentially expressed miRNAs predicted to target protein-coding genes involved in pathologies of atherosclerotic aetiology[20]. Moreover, further studies will be necessary to confirm these promising results in the genomic field. In addition, Kremers et al [21] underlined the usefulness of high-sensitivity CRP, neutrophil-lymphocyte ratio, NT-proBNP, and high-sensitivity cTnT, that seemed to be more feasible also in common laboratories, be-cause it implies only blood samples. Combining these markers for individual risk stratification may lead to improved treatment choices and increased effectiveness of current treatment strategies.
People from low-income countries seem to have a higher prevalence of intermittent claudication (IC) and CLTI, due to the major exposure to all these risk factors.”
- Describe the economic burden of PAD and PAD with CTLI.
Response: thank you for your comment
We expanded this topic as follows:
“… worldwide social and economic impact of CLTI in the modern age. As already mentioned, patients with PAD may present a wide range of symptoms ranging from claudication up to extensive necrosis or gangrene. Most of them require hospitalization for surgical or endovascular interventions, others need frequent outpatient visits to assess the stability or progression of the disease, or need dressing cycles for non-healing ulcers. It has been calculated that the rate of hospitalizations for PAD during 2014 in the USA was 89.5/100,000, with 137,050 (or 45%) of these having a high grade disease. For a mean hospital stay of 5 days the cost was $15,755, resulting in an annual cost burden for hospitalization of patients with PAD of ∼$6.31 billion [23]. The direct costs associated with PAD are higher than those for cardiovascular disease, because of the polyvascular feature of the pathology and the higher number of annual cardiovascular events and hospitalization rates. In addition to direct costs, PAD may lead to large morbidity- and mortality-related productivity costs. The 2010 National Health and Wellness Surveys developed in the USA and Europe, reported significant impairment in work in patients with PAD: in particular absenteeism, presenteeism, overall work productivity loss, and activity impairment [24]. This fact underlines the need for strict risk factors ‘control and a correct use of guidelines recommended drugs.”
- Patients apparently similar in overall risk, comorbidities, and clinical characteristics have significant differences in outcomes after the revascularization procedure. Briefly discuss the current limitations affecting reliable population risk stratification.
Response: thank you for your comment
We discussed current limitations of population risk stratification as follows:
“Despite the large improvement in risk factors’ control and medical treatment, the number of PAD patients who need a revascularization continues to be high. The estimation of life expectancy and operative risk plays a central role in evidence based revascularization (EBR). A lot of models have been developed during years to stratify the risk of these patients. The existing risk models have demonstrated modest predictive abilities; indeed, patients apparently similar in overall risk, comorbidities, and clinical features have shown significant differences in terms of outcomes after the revascularization. This lack of predictive ability is due to the heterogeneous nature itself of these models regarding predictor variables and outcomes assessed. Firstly, the inclusion of endovascular therapies is not uniform as well as the evaluation of the severity of foot necrosis or the adequacy medical risk-factor management and other risk factors. Secondly, all of the scoring systems have difficulties to be generalized to the entire population, due to the lack of rigorous external validation. Then, another bias is due to indication, because many of the derivation sets included only infrainguinal bypasses or angioplasties, but not both. Eventually, with acquisition of newer intraoperative and postoperative predictors of outcomes in CLI, these systems become progressively complex [25].”
- Page 2, line 69: correct the citation number (14). That paper (narrative review and not a metanalysis) is an example of original contribution (impact of dietary risk factors, eating behaviours and nutritional habits on PAD and outcomes of PAD).
Please correct any other mistakes in the bibliography and reference association.
Response: thank you for your comment
Page 2, line 69: we corrected the association between bibliography and reference to “[15]”. We also changed “a recent metanalysis” to “a recent narrative review explained …”
- Page 2, line 80-82: the latest CTLI guidance is from 2019. Did you refer to a specific topic covered in the 2015 guidelines? Otherwise correct.
Response: thank you for your comment
Page 2, line 80-82: we referred to the 2019 Global Vascular Guidelines reporting 4-year mortality rate, so we correct “latest Guidelines” to “Global Vascular Guidelines”.
Chapter 2, Diagnosis:
- Page 2, sub-chapter 2.1. Physical examination, line 94-95: pulse palpation may be useful as bed-side approach to suspect PAD. Furthermore, pulse palpation may not be effective in diagnosing and assessing the severity of PAD.
Response: thank you for your comment
We modified the sentence to clarify the correct meaning as follows:
“… Palpation of lower limb pulses from the groin to the foot (femoral, popliteal, pedis, posterior tibial) is useful as a bed-side approach to suspect PAD. However, pulse palpation may not be effective in diagnosing and assessing the severity of PAD.”
- Page 4, sub-chapter 2.3. WIfI classification system, line 135-139: the comparison with TNM is not really useful for the purpose of the review and the reference added at the end of the phrase doesn't mention any of the similarities reported.
Response: thank you for your comment
We modified the text by deleting this comparison and the reference related as well.
“… three main factors: wound, ischemia, and foot infection. First of all, the close similarity between WIfI classification system and Tumor-Node-Metastasis (TNM)-based staging of cancers should be taken into consider-ation. TNM classification system is predictive for good or poor outcomes in oncological patients, whilst WIfI score is useful to correlate the grade of the disease to the risk of major amputation, wound healing, and mortality [8].”
- Diagnosis is critical to the management of PAD. More importantly, it appears to be the patients' dynamic assessment, functional status, and limitations to daily activities that are directly associated with outcomes. Measurements of lower limb muscle mass, walking tests, and quality of life questionnaires are promising tools for improving decision-making and risk stratification of patients with PAD.
Response: thank you for your comment
We added your relevant comment as a part of the diagnosis section as follows:
“… physical examination, noninvasive hemodynamic tests, and imaging.
Diagnosis is critical to the management of PAD. More importantly, it appears to be the patients' dynamic assessment, functional status, and limitations to daily activities that are directly associated with outcomes. Measurements of lower limb muscle mass, walking tests, and quality of life questionnaires are promising tools for improving decision-making and risk stratification of patients with PAD.”
Chapter 3, The Global Limb Anatomic Staging System (GLASS):
- sub-chapter 3.2: Limb staging, page 6, line 213-214: It could be useful to adapt the graphic referred to in the paper and add it so the reader can have a quick overview
Response: thank you for your comment
Sub-chapter 3.2: Limb staging, page 6, line 213-214: we reproduced the graphic of GLOBAL Guidelines and inserted it during the text as follows:
“… its meaning can be easily understood looking at the graphic reported in the Global Vascular Guidelines, which suggests high benefits for the revascularization in selected categories of patients (Fig 2).”

Fig. 2: reproduction of the graphic present in GLOBAL Guidelines which illustrates the benefit of performing revascularization in CLTI
- line 185: what do you mean by saying “It is usually the most diseased crural artery providing runoff to the foot”? Please, clarify.
Response: thank you for your comment
Page 6, line 185:we clarified this concept correcting the mistake as follows:
“… the groin to the foot through a target arterial pathway (TAP). This latter is usually selected because is the least diseased (or the more suitable) crural artery providing runoff to the foot [13].”
Chapter 4, Evidence-Based Revascularization
- Line 263: common femoral artery disease is another indication for surgery as first recommended treatment.
Response: thank you for your comment
Line 263: we modified the sentence as suggested as follows:
“However, the treatment of choice for inflow disease located on femoral bifurcation as well as common femoral artery disease properly, still remains endarterectomy, which has shown to have low perioperative complications, and long-term durability.”
- The revascularization procedure itself is an important risk factor for MACE and MALE in PAD population. Therefore, evidence-based revascularization also means identifying all those cases where revascularization would only provide an increased risk of mortality and morbidity rather than a benefit. In fact, “unnecessary revascularization” and the support of risk-modifying therapy appear to be the real future for effective PAD management. Please discuss.
Response: thank you for your comment
We expanded the topic as follows:
“... to achieve an in-line flow to the foot the “plantar loop technique” is nowadays well-described and widely applied.
Despite these numerous advances in revascularization techniques and anesthetic drugs, surgical or endovascular revascularization may not be feasible in some patients, even if technically possible, because of significant comorbidities and reduced life expectancy. Patients with PAD are at increased risk for MACE and MALE especially in the perioperative period. Furthermore, these events are not limited to patients with the most severe manifestations of PAD, but are also seen in patients with less severe ones [73]. Therefore, the goal of EBR is also identifying those patients where revascularization would only provide an increased risk for MACE and MALE rather than a real benefit. Therapeutic choice should avoid unnecessary revascularizations, and should be based on the patients as a whole rather than aiming only at limb salvage. In this scenery, focusing on improved risk-modifying therapy might be the key for the future management of PAD.”
- Weissler E H, Wang Y, Gales J M, Feldman D N, Arya S , Secemsky E A, Aronow H D, Hawkins B M, Gutierrez J A, Patel M R, Curtis J P, Schuyler Jones W, Swaminathan R V. Cardiovascular and Limb Events Following Endovascular Revascularization Among Patients ≥65 Years Old: An American College of Cardiology PVI Registry Analysis. J Am Heart Assoc 2022, 20;11(12):e024279.
- Vascular Team and multidisciplinary evaluation of cases is recommended for the most challenging patients. Please discuss the importance of shared decision making and the effects of multidisciplinary management of PAD patients.
Response: thank you for your comment
We expanded this topic as follows:
“4.1 Multidisciplinary team management
A single specialist does not possess all necessary skills to manage complex patients with multiple comorbidities. For this reason it is useful to create a team of specialists with the skills required. An intermediate model of multidisciplinary team dedicated to the management of CLTI patients should include at least vascular surgeon, endocrinologist, interventionalist, orthopedic surgeon, podiatric surgeon, diabetic and wound nurse, physical therapist, diabetes educator and nutritionist [13]. Therefore, multidisciplinary treatment of CLTI is often defined as the combination of bypass surgery, endovascular treatment, wound healing, or rehabilitation therapy that is performed for the purpose of saving limbs and lives. These intensive and complementary therapies performed from various specialists are essential for successfully treating these patients. They can be evaluated from multiple perspectives, and the best treatment could have been selected after a collective decision and opinions’ sharing. The impact of multidisciplinary teams has been well established: they can improve processes, time to intervention, and outcomes. Indeed, Mii et al demonstrated that aggressive wound care by a multidisciplinary dedicated team shortened the time to wound healing and increased the rate of wound healing within 1 year [74]. Similarly, Zayed et al showed that a multidisciplinary approach improves the limb salvage rate of high-risk CLTI patients [75].
In summary, the best way to treat CLTI patients, especially those with multiple comorbidities, is offering an integrated multidisciplinary approach able to manage all aspects of the disease.
- Mii S, Tanaka K, Kyuragi R, Ishimura H, Yasukawa S, Guntani A, Kawakubo E. Aggressive Wound Care by a Multidisciplinary Team Improves Wound Healing after Infrainguinal Bypass in Patients with Critical Limb Ischemia. Ann Vasc Surg 2017;41:196-204.
- Zayed H, Halawa M, Maillardet L, Sidhu P S, Edmonds M, Rashid H. Improving limb salvage rate in diabetic patients with critical leg ischaemia using a multidisciplinary approach. Int J Clin Pract 2009;63(6):855-8.
Conclusion:
- The conclusion may appear vague as it does not remark the importance of “Evidence-Based Revascularization” and “Necessary Revascularization”. Rewrite the conclusion including the revisions suggested in this peer review.
Response: thank you for your comment
We rewrote the conclusion including the revisions suggested as follows:
“Conclusions
CLTI not only is one of the most widespread and challenging diseases for vascular surgeons but also one of the most high impact burdens on the economic system. Global Vascular Guidelines have improved the way to manage CLTI patients through an evidence based revascularization (EBR). Revascularization itself represents a non-negligible risk for MACE and MALE, reason why an accurate stratification considering all the individual risk factors, both clinical and non-clinical, and the new idea of “residual risk” for the patient, is fundamental in the management of this pathology. In fact, what is detectable from the current literature is that patients apparently similar in overall risk, comorbidities, and clinical characteristics have significant differences in outcomes after the revascularization procedure. The aim of EBR is to only perform procedures where the risk-benefit ratio is well balanced, so as to avoid overtreatment and unnecessary revascularizations. For sure, in the most challenging patients Vascular Team and multidisciplinary evaluation of cases is strongly recommended together with the expertise of the surgeon. For what concerns patients with an average risk the BEST-CLI trial established that open surgical peripheral vascular surgery is still alive, guaranteeing better outcomes rather than the less invasive endovascular approach.”
Minor Comments
- Overall grammar (especially use of prepositions) should be revised
Response: thank you for your comment
We reviewed the manuscript trying to correctly use prepositions as suggested.
Examples:
Page 1, line 45-46: changed from “associated to„ to “associated with„
Page 2, line 62: changed from“PAD is increasing during last years„ to “PAD has been increasing during last years„
Page 2, line 64: changed from “ people has PAD „ to “ people have PAD „
Page 2, line 66: changed from “dramatically with the age„ to “dramatically with age „
Page 3, line 101: changed from “ especially in patient„ to “ especially in patients„
Page 5, line 146: changed from “gives to the physicians „ to “gives to physicians „
Page 5, line 168: changed from“only in few selected cases „ to “ only in a few selected cases „
Page 5, line 178: changed from“ CLTI has been not standardized „ to “CLTI has not been standardized„
Page 6, line 214: changed from“ to relief pain „ to “to relieve pain„
Page 7, line 225: changed from“ unless clinical status allowed „ to “ unless clinical status allows „
Page 7, line 226: changed from“ could be benefit „ to “ could be beneficial„
Page 7, line 226-227: changed from“ in case of both advanced tissue loss or infection „ to “ in case of either advanced tissue loss or infection „
Page 7, line 252-253: changed from“ in endovascular field „ to “ in the endovascular field „
Page 9, line 298: changed from“ small saphenous vein„ to “ small saphenous veins „
Page 10, line 326: changed from“ who need „ to “who needed„
Page 10, line 329: changed from“ adverse limb event „ to “adverse limb events„
Page 11, line 335: changed from “to surgical group„ to “to the surgical group„
Page 11, line 339: changed from “ those in endovascular group „ to “those in the endovascular group„
Page 11, line 349-350: changed from “ surgical and endovascular group „ to “surgical and endovascular groups„
Page 11, line 370: changed from “in endovascular field„ to “in the endovascular field„
Page 11, line 378: changed from “ In the last years „ to “In the last few years„
Page 12, line 381: changed from“heterogenous„ to “heterogeneous„
- There is excessive use of passive tenses which overcomplicates the reading
Response: thank you for your comment
We reviewed the manuscript trying to avoid excessive use of passive tenses as much as possible, but compatible with the sentences’ structure.
Page 2, lines 49-51: changed from“in 2019 the “Global Vascular Guidelines on the Management of Chronic Limb-Threatening Ischemia” have been released, where ... „ to “in 2019 the European Society of Vascular Surgery (ESVS) has realised the “Global Vascular Guidelines on the Management of Chronic Limb-Threatening Ischemia”, where ... „
Page 4, line 130-131: changed from“ Therefore, the combination of all these tests should be considered the best way to define the grade of PAD and CLTI [22] „ to“ Therefore, the best way to define the grade of PAD and CLTI should be the combination of all these tests [22] „
Page 5,lines 163-165: changed from“ when a complete overview of the vascular bed is needed for complex invasive interventions. „ to“ especially when complex invasive interventions require a complete overview of the vascular bed. „
Page 6, lines 197-198: changed from “Deficits of inflow must be corrected for a successful and durable revascularization „ to “A successful and durable revascularization should correct deficits of inflow „
- Page 6, line 217: the wording is not clear
Response: thank you for your comment
To clarify the concept we changed the sentence as follows:
From “The assessment of the limb stage plays a central role in the GLASS classification due to the wide spectrum of CLTI clinical presentations“ to “Due to the wide spectrum of CLTI clinical presentations, the assessment of the limb stage with the GLASS classification system plays a central role.“
- Page 10, line 349-350: missing reference
Response: thank you for your comment
We added the reference number “[76]”about BEST-CLI trial.
- Farber A, Menard MT, Conte MS, et al. Surgery or Endovascular Therapy for Chronic Limb-Threatening Ischemia. N Engl J Med 2022, 387, 2305-2316.
- Page 10, line 364: Reference number 64 is not listed in the “References” section
Response: thank you for your comment
We changed the number of reference for page 10, line 364 regarding no-option CLTI patients from “[63]” to “[77]” after addition of some other references. Now these references are reported as follows:
- 77. Troisi N, D'Oria M, Fernandes E Fernandes J, et al. International Union of Angiology Position Statement on no-option chronic limb threatening ischemia. Int Angiol 2022, 41, 382-404.
Reference “[64]” has been changed to “[78]” concerning therapeutic efficacy of autologous non-mobilized enriched circulating endothelial progenitors in patients with CLI and listed in the References.
- 78. Liotta F, Annunziato F, Castellani S, et al. Therapeutic Efficacy of Autologous Non-Mobilized Enriched Circulating Endo- 560 thelial Progenitors in Patients With Critical Limb Ischemia - The SCELTA Trial. Circ J 2018, 82, 1688-1698.
- Page 11, line 421: reference number 13 is overcited throughout the paper
Response: thank you for your comment
We delated, as suggested, the overcited reference 13 from the passages where it could be considered superfluous as follows:
Page 2, lines 67-68: “In addition, 67 PAD seems to be more prevalent among black individuals than among whites [13].“
Page 2, lines 81-83: “Furthermore, Global Vascular Guidelines reported a 4-year mortality rate of 18.9% for patients in Rutherford class 1-3, 37.7% for patients in Rutherford class 4, 52.2% for patients in Rutherford class 5, and 63.5% for patients in Rutherford class 6 [13].”
Page 6, lines 206-207: “The objectives of revascularization in CLTI patients are well-known: first, relief of 206 pain; second, wound healing; third, preservation of the limb’s function [13].”
Page 6, lines 212-214: “Revascularization as a palliative treatment should be considered only to improve the inflow for a subsequent amputation, and to relief pain [13, 45].”
Page 8, lines 266-267: “Deficit of inflow is defined in Global Vascular Guidelines [13].”